# Chronic Voluntary Alcohol Consumption Alters Promoter Methylation and Expression of *Fgf-2* and *Fgfr1*

**DOI:** 10.3390/ijms24043336

**Published:** 2023-02-07

**Authors:** Leonie Herburg, Mathias Rhein, Sabrina Kubinski, Ekaterini Kefalakes, Matar Levin Greenwald, Simona Gielman, Segev Barak, Helge Frieling, Claudia Grothe

**Affiliations:** 1Institute of Neuroanatomy and Cell Biology, Hannover Medical School, 30625 Hannover, Germany; 2Center for Systems Neuroscience (ZSN), 30559 Hannover, Germany; 3Department of Psychiatry, Social Psychiatry and Psychotherapy, Hannover Medical School, 30625 Hannover, Germany; 4School of Psychological Sciences, Tel Aviv University, Tel Aviv 69978, Israel; 5Department of Neurobiology, The George S. Wise Faculty of Life Sciences, Tel Aviv University, Tel Aviv 69978, Israel; 6Sagol School of Neuroscience, Tel Aviv University, Tel Aviv 69978, Israel

**Keywords:** FGF-2, FGFR1, alcohol, addiction, epigenetic alteration, mesolimbic system

## Abstract

Alcohol abuse accounts for 3.3 million deaths annually, rendering it a global health issue. Recently, fibroblast growth factor 2 (FGF-2) and its target, fibroblast growth factor receptor 1 (FGFR1), were discovered to positively regulate alcohol-drinking behaviors in mice. We tested whether alcohol intake and withdrawal alter DNA methylation of *Fgf-2* and *Fgfr1* and if there is a correlation regarding mRNA expression of these genes. Blood and brain tissues of mice receiving alcohol intermittently over a six-week period were analyzed using direct bisulfite sequencing and qRT-PCR analysis. Assessment of *Fgf-2* and *Fgfr1* promoter methylation revealed changes in the methylation of cytosines in the alcohol group compared with the control group. Moreover, we showed that the altered cytosines coincided with binding motives of several transcription factors. We also found that *Fgf-2* and *Fgfr1* gene expression was significantly decreased in alcohol-receiving mice compared with control littermates, and that this effect was specifically detected in the dorsomedial striatum, a brain region involved in the circuitry of the reward system. Overall, our data showed alcohol-induced alterations in both mRNA expression and methylation pattern of *Fgf-2* and *Fgfr1*. Furthermore, these alterations showed a reward system regional specificity, therefore, resembling potential targets for future pharmacological interventions.

## 1. Introduction

According to the World Health Organization, 10–15% of the population worldwide suffer from harmful alcohol use. However, specific pharmacotherapeutic approaches are limited to date [1,2]. Alcohol addiction is thought to depend on neural adaptations in the brain that result from chronic alcohol consumption [3,4]. These alcohol-induced neural adaptations occur mainly in circuits of the reward system, which is composed of the mesocorticolimbic dopaminergic (DAergic) pathway, projecting from the ventral tegmental area (VTA) to structures closely related to the limbic system, most prominently the nucleus accumbens (NAc), hippocampus, and prefrontal cortex (PFC) [5,6,7] as well as the nigrostriatal DAergic projections from the substantia nigra pars compacta (SNpc) to the dorsal striatum [6,8]. The nigrostriatal system plays a pivotal role in developing goal-directed behaviors toward the habitual and compulsive nature of drug addiction [6]. Alcohol addiction behaviors are strongly linked to the reward system as its consumption is associated with the activation of DAergic neurons in the VTA and the subsequent release of dopamine, one of the main neurotransmitters related to reward and pleasure [9,10].

Fibroblast growth factor 2 (FGF-2), a multifunctional factor with neurotrophic activities, is known to have a regulatory role in the development and maintenance of DAergic neurons and hence, nigrostriatal pathways [11,12,13,14]. In particular, *Fgf-2* deficient mice display abnormalities in the DAergic system, including a developmental hyperplasia of the SNpc [12,15] accompanied with an increased striatal volume leading to increased DAergic transmission [16,17]. FGF-2 is ubiquitously expressed in neural cells throughout the central nervous system, specifically in the mesolimbic and nigrostriatal circuitries, the hippocampus, and the frontal cortex, where it is involved in cell proliferation, differentiation, proliferation, survival, and migration [18,19,20]. Its pleiotropic effects are mediated via binding, thereby activating high-affinity FGF receptors (FGFRs), mainly FGFR1 [21,22]. Extracellular binding of FGF-2 to FGFR1 induces activation of several intracellular signaling pathways, namely, mitogen-activated protein kinase (MAPK)/extracellular signal-regulated kinase (ERK) [23,24], phospholipase C gamma (PLCγ), and phosphatidylinositol-3-kinase (PI3K)/protein kinase B (PKB; also known as AKT) [25]. 

Recently, it has been shown that both Fgf-2 and Fgfr1 are involved in regulating alcohol consumption [26]. Acute alcohol treatment upregulated *Fgf-2* and *Fgfr1* mRNA in the dorsal striatum of mice, while chronic consumption limited those alterations to the dorsomedial striatum (DMS) [26,27]. Moreover, infusion of recombinant FGF-2 directly into the DMS increased both alcohol intake and preference, whereas inactivation of endogenous FGF-2 [27] or inhibition of FGFR1 [26] suppressed alcohol intake. Those findings suggest a pivotal role of the FGF-2/FGFR1 system in alcohol-drinking behaviors, and specifically within the dorsal striatum. However, the exact mechanisms of how the FGF-2/FGFR1 system influences alcohol consumption and addictive behavior are unknown.

Latest research highlights the importance not only of gene expression itself but also of epigenetic regulation of genes encoding for neurotrophic factors, including nerve growth factor (NGF) and brain-derived neurotrophic factor (BDNF), in alcohol addiction [28]. It is suggested that epigenetic regulation (e.g., DNA methylation) of these growth factors may contribute to the development of alcohol-addictive behaviors [29,30,31]. Nevertheless, less is known about the underlying mechanisms of epigenetic regulation of either *Fgf-2* or its main receptor, *Fgfr1*, in the context of alcohol addiction.

Thus, we hypothesized that chronic alcohol consumption alters the methylation pattern of *Fgf-2* and its receptor *Fgfr1*. Furthermore, this change might be associated with a differential mRNA expression of the aforementioned genes. Indeed, we found that chronic voluntary consumption of alcohol led to both brain region-specific alterations in DNA methylation and mRNA expression of *Fgf-2* and *Fgfr1*. Thus, within this study we showed that FGF-2 and its major target, FGFR1, play important roles in alcohol drinking behaviors and, therefore, provide potential targets for future pharmacological modifications. 

## 2. Results

### 2.1. Voluntary Alcohol Drinking Alters Promoter Methylation and Expression of Fgf-2 in a Brain Region-Specific Manner

First, we examined the drinking behavior of male mice that were trained to consume 20% (*v*/*v*) alcohol in intermittent access to 20% alcohol in a two-bottle choice (IA2BC) drinking procedure [32,33] (Appendix A). During six weeks of intermittent access to 20% alcohol, mice exhibited stable consumption levels with an average of 9 g/kg/24 h (Appendix A). 

Next, we analyzed whether voluntary alcohol drinking alters DNA methylation of the *Fgf-2* gene and, if so, whether these changes are restricted to distinct brain areas of the reward system. Blood and six brain regions, including PFC, NAc, dorsolateral striatum (DLS), DMS, VTA, and SNc, were analyzed regarding changes in their methylation patterns. We found that repeated cycles of voluntary alcohol drinking were associated with an altered methylation pattern of the *Fgf-2* promoter independent of the analyzed brain regions (Figure 1a, Appendix A). Specifically, alcohol consumption led to significant hypomethylation in seven CpG positions within the *Fgf-2* promoter, most prominently in DLS, DMS, and NAc (Figure 1b). In contrast, alcohol-induced hypermethylation exclusively occurred at CpG positions downstream of the start codon within exon 1, especially in the PFC, NAc, DLS, and VTA (Figure 1b). While chronic alcohol consumption in NAc and DLS led to both hypo- and hypermethylation, only hypermethylated CpG positions were observed in the PFC and VTA areas (Figure 1b). 

Since a promoter exhibits many binding sites for transcription factors that are involved in the subsequent transcriptional regulation of the gene, in silico analysis for potential binding motifs overlapping with differential methylated CpG positions was further performed. In silico analysis of transcriptional sites revealed several transcription factors of *Fgf-2* that may be affected by methylation (Appendix A). Factors whose binding motif overlapped with the respective CpG position are shown in Table 1.

In order to examine whether changes in the epigenetic regulation coincide with a differential gene expression, we examined the mRNA levels of *Fgf-2* in the respective brain areas (Figure 2). We found that *Fgf-2* expression was decreased in the DMS and observed a trend for the VTA (*p* = 0.08) of mice consuming alcohol compared with water-drinking control animals. However, alterations in the expression of the *Fgf-2* gene were only statistically significant in the DMS (Figure 2).

Treatment effects were compared in a linear mixed model by including tissue and treatment as factors and expression data as a covariate. For *Fgf-2,* we observed that the effects of tissue, treatment group, and expression level (F = 22.559; *p* = 6.8 × 10^−46^) increased the statistical significance compared with only combining tissue and treatment group (F = 25.110; *p* = 3.9 × 10^−25^) (Appendix A). Expression levels did not only correlate significantly by themselves (tissue: F = 56.890, *p* = 5.2 × 10^−58^; treatment group: F = 76.360, *p* = 3 × 10^−18^; expression: F = 9.844, *p* = 0.002), but also increased the effect on promoter methylation synergistically (Appendix A). This further strengthened the initial evidence of the influence of alcohol consumption on *Fgf-2* promoter methylation.

### 2.2. Effects of Chronic Alcohol Consumption of Fgfr1 Promotor Methylation and Expression

Since FGFR1 is the main target of FGF-2 in the nervous system [34], we also analyzed its promoter methylation and expression levels in the aforementioned brain circuits. Mean methylation of the *Fgfr1* promoter did not significantly differ between alcohol and control groups (Appendix A). However, in the PFC (−270 bp) and DLS (−361 bp), one CpG position was hypermethylated in alcohol-drinking animals compared with the control animals (Figure 3, Appendix A). In silico analysis of differentially methylated CpG positions for putative transcription factor binding motifs revealed a number of factors that are potentially affected in their binding by methylation. Identified transcription factors with potentially affected binding motifs were SP1 (CpG −361 bp), ZNF460 (CpG −361 bp), and NFE2 (CpG −270 bp) (Appendix A).

Similar to FGF-2, we also tested whether chronic alcohol consumption affects mRNA levels of *Fgfr1* in the brain areas of interest. We observed that *Fgfr1* mRNA levels were downregulated in the DMS after six weeks of alcohol consumption compared with the control group (Figure 4). In contrast, alcohol had no effect on *Fgfr1* expression in other brain areas (Figure 4). 

MLM analysis of the *Fgfr1* promoter methylation as a dependent variable did not reveal any significant interactions between methylation and tissue, treatment, or expression data (Appendix A). 

## 3. Discussion

Our results indicate that alcohol consumption affects the methylation pattern of both *Fgf-2* and *Fgfr1* and suggest that this plays a possible role in regulating transcription. 

Specifically, by using a well-established mouse model for alcohol consumption, our main findings were: (a) stable alcohol intake of male mice exposed to the IA2BC drinking procedure, (b) chronic consumption of alcohol led to both hypo- and hypermethylation of the *Fgf-2* gene in a brain region-specific manner, (c) CpG positions methylated differently due to alcohol intake coincided with binding motifs of important transcription factors, (d) mRNA expression levels of *Fgf-2* and *Fgfr1* decreased following chronic alcohol consumption exclusively in the DMS, and (e) treatment group, brain region, and expression levels of *Fgf-2* were all statistically significant parameters which synergistically affected methylation of the *Fgf-2* and *Fgfr1* genes. 

We observed that male C57BL/6J mice that were offered intermittent access to 20% ethanol and water voluntarily consumed stable amounts of ethanol with an average of 9g/kg/24 h (Appendix A). These results are consistent with other studies showing that male C57BL/6J mice consumed similar levels (10 g/kg/24 h) when offered 20% of alcohol [33]. Furthermore, mice lacking FGF-2 consumed significantly lower amounts of ethanol compared with wild-type mice [33]. Altogether, this leads to the assumption that multiple cycles of heavy drinking bouts and withdrawal periods lead to neuroadaptations in certain brain areas that induce ethanol dependence and addiction [35,36].

Abnormal methylation of cytosines in the promoter region of genes can be caused by exogenous stimuli such as alcohol intake and often correlate with a decreased transcription of the respective gene [37]. We observed that prolonged alcohol consumption led to long-lasting changes in *Fgf-2* and *Fgfr1* DNA methylation in brain reward circuits of male mice (Figure 1 and Figure 3). Regarding *Fgf-2*, we identified 70 CpG positions in the region of the transcription start site (Appendix A). Specifically, we showed that alcohol-drinking mice exhibited lower DNA *Fgf-2* methylation levels upstream CpG +552 and higher levels downstream, independent of a specific brain region. Accordingly, CpG +552, which is located directly downstream of the ATG start codon, can be considered as the point where alcohol-induced effects on *Fgf-2* promoter methylation are reversed. Most significant changes regarding methylation were found in the VTA. Interestingly, no alterations with respect to *Fgf-2* methylation could be observed in the SNc of alcohol-drinking mice (Appendix A).

Recent research emphasizes the fact that environmental factors such as alcohol consumption have a profound impact on gene expression mediated via epigenetic mechanisms [30]. Ciafré and colleagues [38] integrated systems approaches, such as those recently used for mapping of novel targets in AUD, as promising approaches to build a set of predictive markers for AUD diagnosis and therapy [39]. Alcohol causes important deregulations at many different levels of action, leading to complex multifactorial homeostatic disorders [40]. These epigenetic alterations, including DNA methylation, are involved in regulating genes without changing the DNA sequence itself [41]. It is assumed that epigenetic regulation of growth factors, such as glial cell line-derived neurotrophic factor (GDNF) and BDNF, may contribute to the development of alcohol dependence [29,30,31]. For example, an association between the pathology of alcohol dependence and alterations in DNA methylation of the *Gdnf* gene was described in rats. Specifically, in the NAc and VTA, a 24-h withdrawal period of alcohol led to a significant decrease in the methylation of the negative regulatory element of *Gdnf* in rats [42]. Moreover, promoter methylation of the *BDNF* gene was significantly increased in alcohol-dependent patients, while alcohol withdrawal resulted in a significant decrease in its methylation [30]. In a mouse study, alcohol-exposed males had lower Bdnf DNA methylation levels in NAc compared with the control mice, while a higher methylation status was detected in the PFC [43]. These findings highlight the impact of individual neurotrophic factors on different brain regions in developing alcohol-addictive behaviors in humans and animals. 

In an independent calculation of a mixed linear model including all relevant factors without cohort stratification, thereby combining expression data with the treatment group and tissue, an increased statistical significance suggests a synergistic contribution of the dependent factors (Appendix A). Changes in methylation in our experiment, therefore, correlated with *Fgf-2* expression, while *Fgfr1* appeared to be regulated by other mechanisms. Future investigations into the interrelationship of methylation and small interfering RNA (siRNA) could shed light on these coregulatory effects.

The abundance of CpG sites near the transcription start site (TSS) suggests a regulatory function of this region, which may be involved in the transcriptional regulation of the *Fgf-2* gene. This hypothesis is further supported by our finding that several CpG sites with differential methylation levels in alcohol-consuming mice comprise transcription factor binding regions which may affect *Fgf-2* gene expression. In silico analysis predicted potential binding sites for the early growth response protein 1 (EGR1), specificity protein 1 (SP1), Yin Yang 1 (YY1), and members of the E2F-transcription factor family (Table 1). EGR1 is a critical transcription factor involved in brain development, long-term neuronal plasticity [44], and higher processes such as learning, memory, or reward [45]. It has already been shown that both the human [46] and rat [47] *Fgf-2* gene exhibit binding sites for EGR1 in the proximal promoter. Additionally, some studies have reported the relevance of SP1 binding sites for the FGF-2 promoter function [48,49]. SP1 is involved in cellular differentiation, cell growth, and apoptosis, and also in chromatin remodeling, mainly in the recruitment of histone acetyltransferases [50]. The transcription factor YY1 is a zinc-finger protein that acts as an activator, a repressor, or an initiator of transcription, depending on the promoter’s context [51]. Since transcription factors have a profound impact on the transcriptional regulation of genes, altered methylation within their putative binding domains could lead to differential expression of the downstream genes, as, for example, *Fgf-2*. Thus, alcohol-induced hypermethylation of the *Fgf-2* promoter in the VTA could be responsible for its reduced mRNA expression in this region. While correlating evidence of significant findings failed for most regions, we observed significant hypomethylation in DMS CpGs +278 and +307, with an associated repressing transcription factor that is activated by alcohol consumption (ATF3) [52,53]. In contrast, this was not the case for the predicted potential activating factors (GABPA, ELF1, GTF2F1, USF1). Since *Fgf-2* mRNA levels in DMS were also significantly lower, a coherent connection between epigenetic regulation and transcription seems plausible (compare DMS in Figure 1b and Figure 2).

Our findings suggest that chronic alcohol consumption affects mRNA expression of *Fgf-2* and *Fgfr1,* especially in the DMS and VTA, which aligns with earlier studies [26,27]. In contrast to previous studies, where mice were offered alcohol over a five-week period, which showed an increase in *Fgf-2* and *Fgfr1* mRNA transcription [26,27], we found a reduction in mRNA levels after six weeks of alcohol consumption (Figure 2 and Figure 4). One critical reason that may contribute to the different effects of ethanol on *Fgf-2* and *Fgfr1* expression could be the timing of tissue collection after the last ethanol drinking session. In contrast with the present study, where tissue sampling occurred after 48 h, Even-Chen and colleagues collected the tissue 24 h after the last session [27]. Moreover, it was already shown for other growth factors that the duration of alcohol exposure seems to have a significant impact on protein and mRNA expression levels [54]. For instance, *Gdnf* mRNA was upregulated following a short drinking period of one week in rats, whereas a reduction was observed after seven weeks [54]. Thus, our results indicate that *Fgf-2* and *Fgfr1* are ethanol-responsive genes whose expression levels not only depend on the duration of alcohol consumption but are also regulated in a brain region-specific manner.

Our current results suggest that pronounced effects of prolonged alcohol consumption are localized to the dorsal striatum. Importantly, this brain region has been shown to play a role in alcohol drinking in rodent models. Specifically, the nigrostriatal system, projecting from the SNc to the dorsal striatum, has been suggested to play a role in habitual and compulsive drinking [55,56]. The dorsal striatum has been implicated as the site of action of different molecules and targets in regulating alcohol drinking [4], including N-methyl-_D_-aspartate (NMDA) receptor [57], the protein tyrosine kinase FYN [58], and BDNF [59,60]. We also recently found that *Fgf-2* and *Fgfr1* regulate alcohol consumption in the two-bottle choice procedure by acting in DMS [26,27]. Thus, our findings suggest that alcohol and FGF-2 interaction is mainly localized to this brain region.

This is the first in vivo study evaluating *Fgf-2* and *Fgfr1* mRNA expression in correlation with its specific promoter methylation under chronic alcohol consumption in mice. We found that *Fgf-2* and its main receptor, *Fgfr1*, play an essential role in the development of alcohol addiction as both promoters show changes in their methylation patterns as well as their gene expression in specific brain areas. Furthermore, this study provides the in-silico identification of alcohol-related transcription factors in the methylation pattern of the *Fgf-2* gene which may serve as potential targets for future pharmacological interventions to prevent alcohol addiction behaviors. 

## 4. Material and Methods 

### 4.1. Ethics Statement

All animal experiments were conducted in strict accordance with the German animal welfare law and were approved by the Lower Saxony State Office for Consumer Protection and Food Safety (LAVES, reference number 33.12-42502-04-18/2977 in accordance with the German Animal Welfare Act).

### 4.2. Animals

The experiments were performed using C57BL/6J wild-type mice of the B6;Cg-Fgf-2tm1Zll strain [61]. Since the genetic background is not standardized, wild-type mice were generated by own breeding. Genotyping was performed as previously described [15].

Male mice (n = 20) were bred at Hannover Medical School (Germany) and kept in the same temperature- and humidity-controlled room on a 14 h light/10 h dark schedule and housed in open cages in groups of four to five with food and water available ad libitum. Starting one-week prior to the drinking experiment, animals were individually housed. The hygienic status was routinely monitored in accordance with the FELASA recommendations [62]. No evidence of infectious agents was revealed except for occasional positive tests for *Rodentibacter pneumotropica* and *Helicobacter* spp.

### 4.3. Intermittent Access to 20% (v/v) Alcohol in Two-Bottle Choice (IA2BC)

The IA2BC procedure was performed as described previously [32,63]. Due to hormonal fluctuations in females and the higher prevalence of alcohol consumption, for the present study only male mice were included. All alcohol sessions were provided in 50 mL falcon tubes with stainless-steel drinking spouts inserted in the front of the cage daily at 11:00. Ethanol from Mallinckrodt Baker, Inc. (Philipsburg, NJ, USA) was diluted to a final concentration of 20% (*v*/*v*) with water. Mice had 24 h sessions of ad libitum access to two bottles per week (one filled with water and one containing 20% (*v*/*v*) alcohol) on Mondays, Wednesdays, and Fridays. During the withdrawal periods, animals had unlimited access to two bottles of water (Appendix A). To prevent side preferences, the position of the solutions was alternated for each drinking session. Water and alcohol bottles were weighed before and after each alcohol-drinking session, with measurements taken to the nearest 0.01 g. The body weight was measured weekly, and consumption levels of water or alcohol were normalized to the body weight of each animal for the respective day. The preference for ethanol over water was calculated by expressing the ethanol intake as a percentage of the total liquid intake [64]. After six weeks of drinking, mice were euthanized 48 h after the last drinking session (11:00), brains were isolated, and the different brain areas were dissected (see below). Data (number, weight, and experimental group) of euthanized mice are listed in Appendix A.

### 4.4. Tissue Processing

Following brain region dissection, the samples were snap frozen in liquid nitrogen, and stored at −80 °C until further use. Frozen tissues were mechanically homogenized with a 25 G cannula, and extraction of DNA, RNA, and total protein amount was performed simultaneously using the AllPrep DNA/RNA/Protein Mini Kit (QIAGEN GmbH, Hilden, Germany) in accordance with the manufacturer’s instruction. Blood extraction and clean-up of genomic DNA was conducted following the NucleoMag Blood 200 µL Kit (Macherey Nagel GmbH & Co. KG, Düren, Germany) on an NxP Biomek liquid handler (Beckman Coulter, Inc., Brea, CA, USA).

### 4.5. Bisulfite Conversion of DNA, PCR Strategy, and Sequencing

After DNA extraction, genomic DNA samples were bisulfite-converted to deaminate unmethylated cytosines to uracils using the EpiTect^®^ 96 Bisulfite Kit (QIAGEN, GmbH) in accordance with the manufacturer’s protocol. Genomic organization of the murine *Fgfr1* and *Fgf-2* gene was obtained using the ENSEMBL genome browser (https://doi.org/10.1093/nar/gkaa942, accessed: 22 February 2021, build 103 by the European Bioinformatics Institute (EMBL-EBI). Primer sequences were designed using the Geneious R11 software (Biomatters Ltd., Auckland, New Zealand) to cover CpG-sites located immediately upstream and within exon 1 of the *Fgf-2* or *Fgfr1* gene and purchased from Metabion (Metabion International AG, Steinkirchen, Germany). To determine melting temperatures, hairpin-formation, and self-dimer formation, the online tool Netprimer was used (http://www.premierbiosoft.com/netprimer/netprlaunch/netprlaunch.html, accessed: 22 February 2021, by Premier Biosoft). Amplification of target sequences was accomplished by polymerase chain reaction (PCR) using HotStarTaq Master Mix Kit (QIAGEN GmbH). Cycler conditions, sequences, and chromosomal position of primers are listed in Appendix A. Amplified PCR products were visualized on a 2% agarose gel and purified using Agencourt AMPure XP magnetic beads (Beckman Coulter, Inc.) in accordance with the manual. A pre-sequencing purification step was conducted using Agencourt CleanSEQ beads (Beckman Coulter GmbH, Krefeld, Germany) and the obtained products were sequenced on an Applied Biosystems 3500xl DNA Analyzer (ABI Life Technologies, Inc., Grand Island, NY, USA). Target product sequencing was performed using the reverse primer with a BigDye^®^ Terminator v3.1 Cycle Sequencing Kit (Applied Biosystems, Inc., Foster City, CA, USA). Data were assessed for quality using Sequence Scanner 2 (ABI Life Technologies, Inc.), and all CpG-related cytosines were quantified with regard to their methylation state using the ESME software package (Epigenetic Sequencing Methylation Analysis Software, Epigenomics AG, Berlin, Germany) [65].

### 4.6. Transcription Factor Analysis

Transcription factor (TF) prediction for significantly different CpG positions between treatment groups was performed with Factorbook service (https://www.factorbook.org/, accessed 24 August 2021, published by the ENCODE Consortium). The relevant cytosine ±7 bases were identified using the TF forward search option, and the ten best matches were assessed for relevance of the cytosine concerning the binding of the transcription factor. If available, information concerning the mode of regulation was acquired from Uniprot (https://www.uniprot.org/, accessed 18 August 2021, published by the European Bioinformatics Institute (EMBL-EBI)), GeneCards (https://www.genecards.org/, accessed 18 August 2021, published by the Weizmann Institute of Science), and JASPAR (https://jaspar.genereg.net/analysis, accessed 18 August 2021, using version 7 of the database published by a consortium of European institutes) websites. An overview of all relevant positions and factors is provided in Appendix A.

### 4.7. Quantitative Reverse Transcription Polymerase Reaction (qRT-PCR)

The expression levels of *Fgf-2* and *Fgfr1* were determined in several brain areas of mice exposed to alcohol or water (control group). A total amount of 1 µg RNA was reversely transcribed to complementary DNA (cDNA) using the iScriptTM cDNA Synthesis Kit in accordance with the manual (Bio-Rad, Cat. #170-8891). For real-time PCR, 25 ng cDNA were mixed with 2 µL diluted primer mix (1.75 µM each forward and reverse primer) and 7 µL POWER SYBR^®^ Green Master Mix (Applied Biosystems, Cat. #4367659). *Fgf-2*, *Fgfr1*, and *Gapdh* primer sequences [15] and melting points of PCR products are listed in Appendix A. Several reference genes (*Gapdh, Ppia, Hprt, 18S*) were tested with *Gapdh,* being the least regulated for our experimental set-up. Thus, *Gapdh* was further used as the reference gene for all subsequent gene analyses. qRT PCR was performed with StepOnePlus^TM^ real-time PCR system and software (Applied Biosystems) with the following protocol. The thermal cycling protocol was as follows: initial denaturation for 10 min at 95 °C, 40 cycles of amplification for 15 s at 95 °C, followed by 1 min at 60 °C. PCR product specificity was determined by melting curve analysis after each cycle. The detection threshold for each primer was set to 0.2. Relative quantification of gene expression was calculated using the ∆∆Ct method.

### 4.8. Statistical Analysis

All statistical analyses were conducted using either SPSS 27 (IBM, Inc., Armonk, NY, USA) or GraphPad Prism 8 (GraphPad, San Diego, CA, USA). 

Alcohol experiments: for all consumption rates, a weekly consumption average was calculated, and data were analyzed with a mixed-model ANOVA, with a between-subjects factor for treatment (alcohol or control), and a within-subjects factor of training week. ANOVA was followed by Fisher LSD post-hoc analysis.

mRNA expression experiments: the mRNA expression of *Fgf-2* and *Fgfr1* was normalized to *Gapdh* expression [27]. Data were analyzed by using Student’s paired *t*-test. 

DNA methylation experiments: methylation levels were parametrically distributed (visual inspection of distribution histograms; see Appendix A). Therefore, we performed group comparisons for Gaussian distributions. Two-tailed *t*-tests were performed for each CpG site to determine significant differences between treatment groups after determining equality of variance by using Levene’s test. Due to the small group sizes and the high number of measurements (Bonferroni correction would require a *p*-value below 0.0007 for FGF-2), none of the significant interactions survived post-hoc testing. 

To test for the interaction between expression levels, treatment group, and promoter methylation, we calculated a mixed linear model (MLM) using the restricted maximum likelihood algorithm with methylation as the dependent variable, treatment and tissue as factors, and qRT-PCR expression data as a covariate. Fixed effects modeling was used to reveal the relation and contribution of these factors on methylation. Multiple measurements (CpG positions per measurement) were accounted for by using the scaled identity algorithm.

Differences were considered significant in cases when the *p*-value ≤ 0.05 and were represented as follows: *: *p*-value ≤ 0.05, **: *p*-value ≤ 0.01, and ***: *p*-value ≤ 0.001.

## 5. Conclusions

To conclude, the role of FGF-2 and its FGFR1 receptor signaling appears to be a significant target for future studies as epigenetic and gene expression alterations are caused by increased alcohol consumption in different areas of neural circuitry.

## Figures and Tables

**Figure 1 ijms-24-03336-f001:**
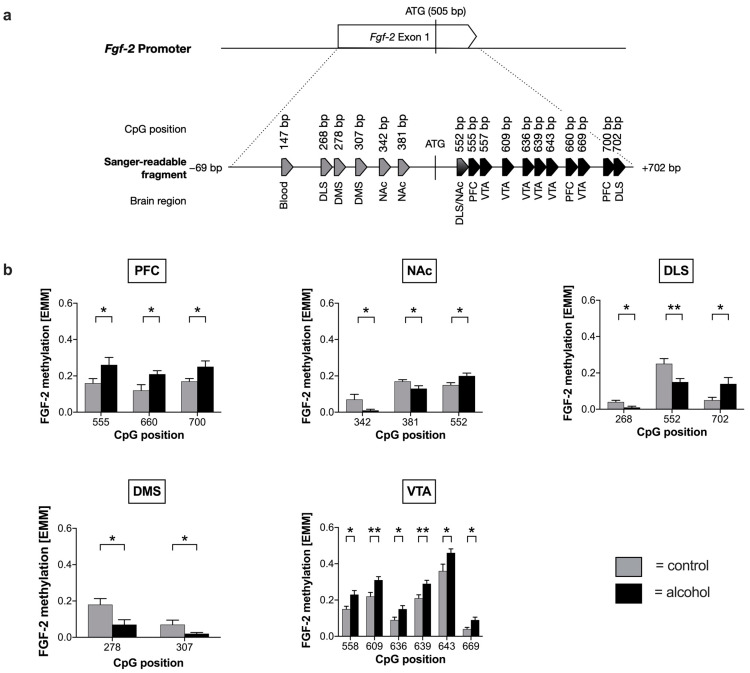
Alterations in the methylation of fibroblast growth factor 2 (*Fgf-2)* after alcohol consumption. (**a**) Schematic representation of significant changes in the methylation pattern of the murine *Fgf-2* promoter and exon 1. Chronic alcohol consumption induced hypomethylation in the *Fgf-2* promoter in blood, dorsolateral striatum (DLS), and dorsomedial striatum (DMS). Increased methylation of CpG positions was observed downstream of ATG within exon 1, being most prominent for prefrontal cortex (PFC), ventral tegmental area (VTA), and DLS. No alterations were observed in the substantia nigra (SNc). We used two-tailed *t*-tests to identify the significantly mapped different CpG positions (total = 17). Arrowheads indicate significant alterations (*p* ≤ 0.05) in methylation rate (gray = hypermethylation in the control group compared with alcohol-drinking mice; black = hypermethylation in the alcohol group compared with water-drinking control mice). (**b**) Methylation rate of significant CpG positions of the *Fgf-2* promoter in PFC, NAc, DMS, DLS, and VTA tissue samples after chronic alcohol exposure. Data are represented as mean ± SEM (n = 10 biological replicates) with a two-tailed Student’s *t*-test, * *p* ≤ 0.05; ** *p* ≤ 0.01 compared with the control group.

**Figure 2 ijms-24-03336-f002:**
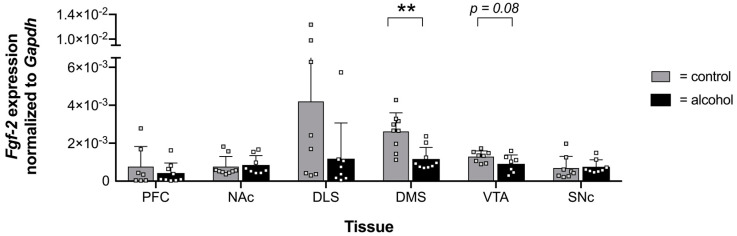
Expression and unified analysis of *Fgf-2*. Expression profiles of the *Fgf-2* gene within six murine brain areas. *Fgf-2* expression was significantly decreased in the dorsomedial striatum (DMS) of mice consuming 20% alcohol for a six-week period compared with the control mice. Data are presented as mean ± SEM with squares depicted as individual data points (n = 7–9 biological replicates) analyzed with Student’s paired *t*-test, ** *p* ≤ 0.01 compared with the control group.

**Figure 3 ijms-24-03336-f003:**
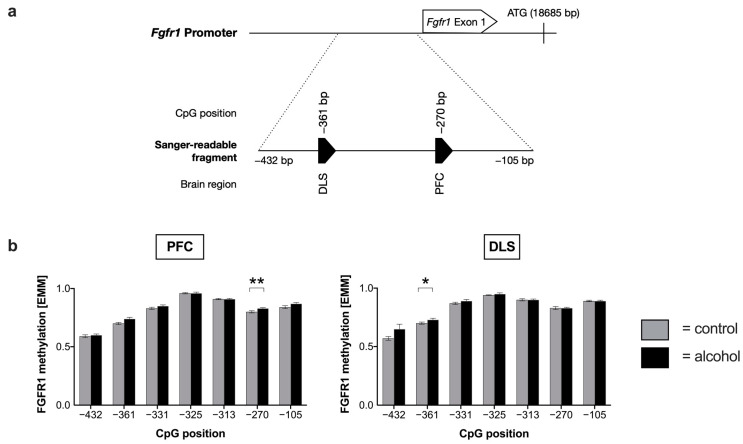
Alterations in fibroblast growth factor receptor 1 (*Fgfr1)* methylation after alcohol consumption. (**a**) Schematic representation of significant changes in the methylation pattern of the murine *Fgfr1* promoter. Chronic alcohol consumption led to hypermethylation of CpG −361 in dorsolateral striatum (DLS) and CpG −270 in prefrontal cortex (PFC). We used bisulfite conversion followed by direct capillary sequencing and two-tailed *t*-tests to identify the significantly mapped different CpG positions (total = 2). Arrowheads indicate significant alterations (*p* ≤ 0.05) in methylation rate (black = hypermethylation in the alcohol group compared with water-drinking control mice). (**b**) Estimated marginal means (EMM) of the methylation rate of all seven examined CpG sites in the *Fgfr1* promoter in PFC and DLS of male mice after chronic alcohol exposure. Data are represented as mean ± SEM (n = 10 biological replicates) with a two-tailed Student’s *t*-test, * *p* ≤ 0.05; ** *p* ≤ 0.01 compared with the control group.

**Figure 4 ijms-24-03336-f004:**
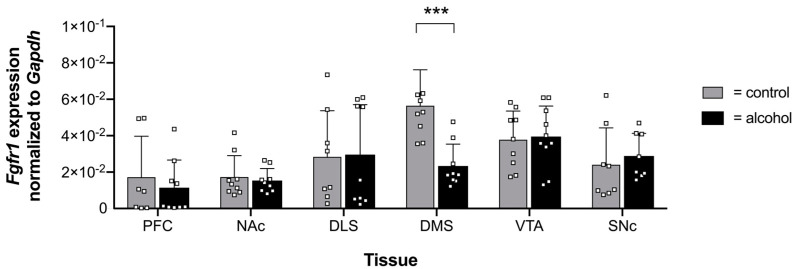
Expression and unified analysis of *Fgfr1*. Expression profiles of the *Fgfr1* gene in six murine brain areas. *Fgfr1* expression was significantly decreased in the dorsomedial striatum (DMS) of mice consuming 20% alcohol for a six-week period compared with the control mice. Data are presented as mean ± SEM with squares depicted as individual data points (n = 7–10 biological replicates) analyzed with Student’s paired *t*-test, *** *p* ≤ 0.001.

**Table 1 ijms-24-03336-t001:** Prediction of transcription factor binding sites within the fibroblast growth factor 2 (*Fgf-2)* promoter. Factorbook was used for binding site prediction (https://www.factorbook.org/, accessed 24 August 2021).

Transcription Factor	Brain Region (CpG Position)	Consensus Sequence
Activating Transcription Factor 3 (ATF3)	DMS (+307 bp) VTA (+639 bp)	5′-GTGACGT[AC][AG]-3′
Early Growth Response 1 (EGR1)	DLS (+268 bp; +702 bp) VTA (+588 bp) PFC (+660 bp; +700 bp)	5′-GCG(T/G)GGGCG-3′
E2F Transcription Factor 1 (E2F1)	DLS (+268 bp) NAc (+342 bp) VTA (+636 bp) PFC (+700 bp)	5′-TTTC[CG]CGC-3′
Specificity Protein 1 (SP1)	DMS (+278 bp)	5′-CCCCGCCCCC-3′
Transcriptional Repressor Protein YY1 (YY1)	DLS/NAc (+552 bp) VTA (+558 bp, +609 bp)	5′-CCGCCATNTT-3′

## Data Availability

The original contributions presented in the study are included in the article/Appendix A; further inquiries can be directed to the corresponding author.

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
