# Peer review of "Chronic Voluntary Alcohol Consumption Alters Promoter Methylation and Expression of Fgf-2 and Fgfr1"

_ijms, 2023, doi:10.3390/ijms24043336_

Round 1

Reviewer 1 Report

In this article, the authors report brain region-specific DNA methylation alteration in the promoter and/or gene body of Fgf-2 and Fgfr1 genes using an alcohol-drinking mouse model. Concerning Fgf-2, statistical modelling shows that alcohol drinking, brain region and gene expression level are significantly associated with CpG methylation observation. However, no simple rule between DNA methylation and gene expression change has been observed, given only in DMS where it shows transcriptional down-regulation of Fgf-2. The authors postulate that DNA methylation changes, especially in the promoter regions may down-regulate gene expression by facilitating the binding of repressing transcription factor ATF3. Thus, the detailed molecular mechanisms are still elusive and more investigations are warranted. In summary, this work is well-designed, -performed and -analyzed and the discussion is informative and relevant.

I have two questions about the DNA methylation assay: how many read-outs of Sanger-sequencing were obtained and used for CpG methylation calculation for one biological sample? And Did the author also calculate the DNA methylation level of non-CG sites in the covered regions?

Author Response

The authors are very grateful for the helpful and constructive comments of the reviewers, which have helped to improve the manuscript significantly. Our responses to the reviewer`s comments are highlighted in italics.

Comments to the authors:

1. how many read-outs of Sanger-sequencing were obtained and used for CpG methylation calculation for one biological sample?

RESPONSE: For the majority of samples, we directly sequenced the bisulfite PCR products without subcloning. In our experience, low variance for groups of analyses, as observed in all reported positions, is an indicator that the sequencing is specific.    

2. Did the author also calculate the DNA methylation level of non-CG sites in the covered regions?

RESPONSE: In the sequencing process, all positions are investigated. The program used for evaluation of the methylation percentage is automatically checking the methylation of non-CpG sites as a means of technical control. Finding non-CpG positions methylated would be a sign of incomplete bisulfite conversion, as these positions usually are not methylated at all

Reviewer 2 Report

The authors aimed to evaluate the effect of subchronic alcohol consumption (20% alcohol intermittent access) in mice susceptible to liver damage (Fgf-2 wild-type mice). The authors provided enough experimental evidence on the alcohol-induced hypo (dorsomedial stratum) and hyper (ventral tegmental area) methylation pattern and mRNA down-regulation of the Fgf-2 promoter in these two brain areas, and that such mechanism is closely related to the alcohol-induced reward effect. The experimental design/execution are impeccable, and the description/discussion of results based on evidence. However, some changes are necessary to improve the scientific soundness and uniqueness of the study.

·         Syntax and grammar should be reviewed by a native English spoken person.

·         Do not forget to describe the meaning of abbreviations the first time they are mentioned.

·         The abstract should be more quantitative than narrative.

·         All figures (both in the body of text and supplementary material) should be provided with a higher resolution (>300 dpi). Also, some changes in the figures could improve the conclusion derived from them. For example, figure 1b seems to be necessary since 1c indicates the point change and Figure 2 could be transformed into a table pointing out only the important sequences.

·         The titles of figures and tables must be differentiated from their corresponding footnotes.

·         In all statistical values, no more than two significant figures should be used.

·         It is recommended to reduce as much as possible the old references (≥10 y) to say 25% or less (currently 53%).

·         It is advisable to include information (for discussion) from new publications on the subject (e.g. DOI: 10.1523/JNEUROSCI.3136-11.2012, 10.1111/acer.13338, 10.1016/j.alcohol.2017.02.357, 10.1016/j.alcohol.2017.02.357, 10.1159/000449486, 10.1139/bcb-2018-0248).

Author Response

The authors are very grateful for the helpful and constructive comments of the reviewers, which have helped to improve the manuscript significantly. Our responses to the reviewer`s comments are highlighted in italics.

Comments to the authors:

1. Syntax and grammar should be reviewed by a native English spoken person.

RESPONSE: A native speaker reviewed our manuscript and improved appropriate parts.

2. Do not forget to describe the meaning of abbreviations the first time they are mentioned.

RESPONSE: We have checked all abbreviations very carefully and corrected corresponding text passages throughout (e.g., Figure 1, line 114-126).

3. The abstract should be more quantitative than narrative.

RESPONSE: We have revised very carefully our abstract.

4. All figures (both in the body of text and supplementary material) should be provided with a higher resolution (>300 dpi).

RESPONSE: We have checked all figures and uploaded them with a higher resolution (>600 dpi) throughout.

5. Some changes in the figures could improve the conclusion derived from them. For example, figure 1b seems to be necessary since 1c indicates the point change and Figure 2 could be transformed into a table pointing out only the important sequences

RESPONSE: Thank you for the suggestions for improvement. Accordingly, we removed figure 1b and instead presented only the significant data in the main text. Moreover, we converted Figure 2 into a table including all important sequences.

6. The titles of figures and tables must be differentiated from their corresponding footnotes.  

RESPONSE: We have checked very carefully the corrected corresponding text passages throughout.

7. In all statistical values, no more than two significant figures should be used.

RESPONSE: We have checked all statistical values very carefully and corrected corresponding passages (e.g., page 6, Figure 2; page 18, Supplemental Table S4).

 8. It is recommended to reduce as much as possible the old references (≥10 y) to say 25% or less (currently 53%).

RESPONSE: The authors would appreciate if we could use our "old" references as these mainly comprise the primary literature or general knowledge. Nevertheless, we added some references recommended by the reviewer (e.g., Ref. 38-40; page 2, line 59).

Reviewer 3 Report

This is an interesting study investigating chronic alcohol consumption and promoter methylation and expression of fibroblast growth factor 2 and receptor 1. The paper is well written and of interest for the journal. I recommend only some minor changes before considering it for publication.

Abstract.

1-It is difficult to separate the different subsections of the abstract. I recommend to include a brief description of AIms and Methods (study design).

Introduction

1- I recommend to add a short paragraph explaining the impact of overall substance use disorders in the brain. It would be a good introduction to the effects of alcohol use.

Why are the results section, and discussion before the section of Material and Methods. I recommend to first explain the methodology of the study, and then the results, discussion and conclusions.

The authors marked that this is the first in vivo study evaluating Fgf-2 and Fgfr1 mRNA expression micer with alcohol consumption. This is a very good strenght of the study. I recommend to include it in the abstract section and highlight in a separate section of Limitations and strenghts (at the end of the discussion).

The main aims of the paper should be clarified at the end of the introduction. Please, expand this part. It can be also separated in a different subsection: 1.1. "Aims".

Author Response

The authors are very grateful for the helpful and constructive comments of the reviewers, which have helped to improve the manuscript significantly. Our responses to the reviewer`s comments are highlighted in italics.

Comments to the authors:

1. Abstract - It is difficult to separate the different subsections of the abstract. I recommend to include a brief description of Aims and Methods (study design).

RESPONSE: We have revised very carefully our abstract

2. Introduction – I recommend to add a short paragraph explaining the impact of overall substance use disorders in the brain. It would be a good introduction to the effects of alcohol use.

RESPONSE: Thank you, for the recommendation. To include overall substance use disorder would be beyond the scope of our manuscript. We decided to focus our study specifically on alcohol use which is complex anyway.

3. Why are the results section, and discussion before the section of Material and Methods. I recommend to first explain the methodology of the study, and then the results, discussion and conclusions.

RESPONSE: The order of the individual sections was dictated by the journal. Therefore, we structured our manuscript this way.

4. The authors marked that this is the first in vivo study evaluating Fgf-2 and Fgfr1 mRNA expression mice with alcohol consumption. This is a very good strenght of the study. I recommend to include it in the abstract section and highlight in a separate section of Limitations and strenghts (at the end of the discussion).

RESPONSE: We have revised very carefully our discussion and abstract according to the recommendations.

5. The main aims of the paper should be clarified at the end of the introduction. Please, expand this part. It can be also separated in a different subsection: 1.1. "Aims".

RESPONSE: We have revised very carefully our introduction section (see p. 3, line 89-95)
